# A Multifaceted Exploration of Status Asthmaticus: A Retrospective Analysis in a Romanian Hospital

**DOI:** 10.3390/jcm13216615

**Published:** 2024-11-04

**Authors:** Adriana Ana Trusculescu, Versavia Maria Ancusa, Camelia Corina Pescaru, Norbert Wellmann, Corneluta Fira-Mladinescu, Cristian Iulian Oancea, Ovidiu Fira-Mladinescu

**Affiliations:** 1Center for Research and Innovation in Personalized Medicine of Respiratory Diseases (CRIPMRD), Pulmology University Clinic, ‘Victor Babes’ University of Medicine and Pharmacy, Eftimie Murgu Square no. 2, 300041 Timisoara, Romania; ana.trusculescu@umft.ro (A.A.T.); pescaru.camelia@umft.ro (C.C.P.); norbert.wellmann@umft.ro (N.W.); oancea@umft.ro (C.I.O.); mladinescu@umft.ro (O.F.-M.); 2Pulmology University Clinic, Clinical Hospital of Infectious Diseases and Pneumophysiology Dr. Victor Babeș Timișoara, Gheorghe Adam Street, no. 13, 300310 Timisoara, Romania; 3Department of Computer and Information Technology, Automation and Computers Faculty, “Politehnica” University of Timis, Vasile Pârvan Blvd, no. 2, 300223 Timisoara, Romania; 4Doctoral School, “Victor Babes” University of Medicine and Pharmacy Timisoara, Eftimie Murgu Square 2, 300041 Timisoara, Romania; 5Hygiene Division, Department of Microbiology, “Victor Babes” University of Medicine and Pharmacy Timisoara, Victor Babes Street, no. 16, 300226 Timisoara, Romania; fira-mladinescu.corneluta@umft.ro; 6Center for Study in Preventive Medicine, “Victor Babes” University of Medicine and Pharmacy Timisoara, Eftimie Murgu Square no. 2, 300041 Timisoara, Romania

**Keywords:** asthma, status asthmaticus, retrospective study, network analysis, geospatial analysis, temporal analysis, comorbidities

## Abstract

**Background**: Status asthmaticus is a severe, life-threatening asthma exacerbation requiring urgent medical intervention. This study aims to examine its epidemiology in Timis County, Romania, over 11 years. **Methods**: A retrospective analysis was conducted using hospital records from 2013 to 2023, focusing on demographic, geospatial, and temporal distributions. Network analysis of the recorded comorbidities was used to identify phenotypic clusters among patients. **Results**: Females and older adults were disproportionately affected. Several triggers and geospatial patterns were identified. Five phenotypic clusters were determined: two in the T2-high endotype, two in T2-low, and a mixed one. **Conclusions**: The findings highlight the need for personalized asthma management strategies and public healthcare interventions in Timiș County, addressing specific demographic and geospatial factors. This study also provides a valuable reference for similar regions.

## 1. Introduction

Asthma is a chronic inflammatory disorder of the airways that produces recurrent episodes with difficulty breathing, often associated with increased hyperactivity to specific stimuli. It is a relatively common disease, impairing almost 4% of the world population [1] and demonstrating an upward trend in prevalence, especially when measured by disability-adjusted life years. The highest percentage of the almost half a million annual asthma-related fatalities comes from patients experiencing slow-onset asthma exacerbations, reflecting inadequate disease control over time, in contrast to sudden-onset exacerbations (20%), which present mostly with clear airways and have a much better outcome [2,3].

The most severe exacerbation, qualifying as a medical emergency, is status asthmaticus. This appears when a patient’s standard asthma treatment with β_2_-agonists and steroids fails to provide relief for a common exacerbation [2]. Its prevalence is 1.23 times higher in adults than in pediatric patients, with its persistent bronchoconstriction leading to hypoxemia, hypercarbia, and secondary respiratory failure [4].

Despite advancements in asthma care, the literature constantly demonstrates heterogeneity in disease presentation and response to treatment. This variability underscores asthma’s complex nature, hindering the development of standard treatment approaches. While the global trend is shifting towards a more personalized approach, ongoing research causes annual updates in the overall asthma treatment and management guide, GINA [5]. Consequently, researchers have explored various class refinements to reach that personalization and address complexity. From a pathophysiological standpoint, two main endotypes are agreed upon, Th2-high and Th2-low, reflecting the different inflammation pathways observed in the airways.

The terms Th2-high and Th2-low refer to distinct asthma phenotypes characterized by the presence or absence of type 2 (T2) inflammation, which is a critical aspect of asthma pathology [6].


**
*Th2-high Asthma*
**


Characteristics:

Patients with Th2-high asthma exhibit elevated levels of T2 cytokines (such as IL-4, IL-5, and IL-13), eosinophils in the blood and sputum, and markers like fractional exhaled nitric oxide (FeNO) above 30 ppb and allergen IgE.

This phenotype is often associated with allergic responses and is more prevalent in individuals with a history of atopy.

Clinical Implications:

Biologic Therapies: Th2-high patients are likely to respond well to targeted biologic therapies that inhibit T2 cytokines, such as anti-IL-5 or anti-IL-4Rα therapies. These treatments can significantly reduce exacerbations and reliance on oral corticosteroids.

Monitoring Biomarkers: Clinicians can use biomarkers like blood eosinophil counts and FeNO levels to identify Th2-high patients, guiding treatment decisions and monitoring responses to therapy.


**
*Th2-low Asthma*
**


Characteristics:

Th2-low asthma is characterized by low levels of T2 inflammation markers. Patients may have normal or low eosinophil counts and lower FeNO levels, indicating a different underlying inflammatory process that may involve neutrophilic inflammation or other non-T2 pathways.

Clinical Implications:

Treatment Challenges: Patients with Th2-low asthma may not respond well to T2-targeted biologics, necessitating alternative treatment strategies. This group often requires a more individualized approach based on their specific inflammatory profile.

Focus on Non-T2 Pathways: Management may include bronchodilators or other therapies that address the non-T2 inflammatory mechanisms present in these patients.

There is an unclear category, so in some cases or instances some patients may have eosinophilic inflammation typical of asthma, while others may present characteristics more aligned with COPD, such as neutrophilic inflammation. The clinical features overlap with both asthma and COPD [7]. The “Dutch hypothesis” tries to answer this question, stating that asthma and BHR predispose to COPD later in life and that asthma, COPD, chronic bronchitis, and emphysema are different expressions of a single airway disease [8]. Furthermore, the presence of these expressions is influenced by host and environmental factors [9].

According to the 2024 Global Initiative for Chronic Obstructive Lung Disease (GOLD) report [10], the term “asthma–COPD overlap” is no longer recommended to describe patients with features of both asthma and COPD. Instead, GOLD advises the following:

Asthma and COPD are Distinct Diseases

GOLD states that asthma and COPD are separate diseases that may share some overlapping characteristics, such as eosinophilia or a degree of reversibility. However, they should be considered distinct clinical entities.

Concurrent Diagnosis of Asthma and COPD

GOLD acknowledges that some patients may be diagnosed with both asthma and COPD concurrently. In such cases, treatment should generally follow asthma guidelines. However, COPD-specific therapies may be needed for some individuals.

Ongoing research attempts to refine these classes, as shown in Table 1, in order to achieve the personalized approach goal.

A common problem with these studies [3,4,5,6,7,8,9,10,11,12,13,14,15,16,17] is their inherent demographic, geographic, and socio-economic heterogeneity. While the literature shows a correlation between asthma and factors within the aforementioned categories, long-term data from relatively stable, homogenous populations remain scarce.

This retrospective study aims to address this gap by examining status asthmaticus cases in a single Romanian county over 11 years. The relatively narrow geographical area (Timis County—8696.7 km^2^) and stable population (~80% the same ethnic group for 30 years) provide unique opportunities for controlled environmental and socio-economic factors, allowing for an exploration of asthma prevalence and severity without confounding influences from diverse populations.

The questions we explore in this study are:Are there temporal patterns in status asthmaticus prevalence over the last 11 years?Do demographic factors such as age and gender influence the risk of status asthmaticus?Are there discernible status asthmaticus associations with known triggers?What are the interactions between status asthmaticus and comorbidities? How do they reflect existing literature phenotypes?

This paper is organized as follows: Section 2, Materials and Methods, presents how data were gathered, analyzed, and processed. The results are in Section 3, divided into the following subsections: statistical descriptions (Section 3.1), time series analysis (Section 3.2), geospatial distribution (Section 3.3), and comorbidities’ complex networks (Section 3.4). Section 3.1 addresses the first and second research questions, while Section 3.2 and Section 3.3 delve into the retrospective analysis of known triggers. Finally, Section 3.4 focuses on the last research question. The discussion is split into three main sections: Statistical Data (Section 4.1), Time and Space Distributions (Section 4.2), and Comorbidities Clusters (Section 4.3). These are each further divided into the evaluation of quantitative data and clinical implications. Since this paper focuses on both empirical data and practical implications, this bifurcation caters to the distinct perspectives of researchers and practitioners, and each subsection can be read on its own. The conclusion aims to summarize the answers to the original questions and provide directions for further studies.

## 2. Materials and Methods

### 2.1. Materials

#### 2.1.1. Study Population

The study population consisted of patients admitted to the Victor Babes University Hospital in Timisoara, Romania, between 1 January 2013 and 30 November 2023. During this time span, we identified 13,695 patients with a primary or secondary diagnosis of asthma according to ICD-10 codes J45.0, J45.1, J45.8, J45.9, and J46. All participants had previously provided written informed consent for their data to be used for research purposes, as all the patients who enter the hospital sign such a waiver.

The population represents 4.69% of the adult population from Timis County. The hospital represents the main respiratory care hub in the county. None of the other university hospitals have dedicated emergency respiratory care facilities and only accept respiratory emergencies in overflow events.

Exclusion criteria were age < 18 years, unknown home address, and address not in Timis County.

#### 2.1.2. Data Collection

Anonymized patient data were extracted from the hospital’s electronic medical records database by the administrative staff and provided to the research team for analysis in an *Excel* format.

#### 2.1.3. Equipment and Software

Data analysis was primarily conducted using the *Python v.3.6* programming language with libraries such as *pandas 2.2.2.* for data manipulation, *NumPy 1.26.4*, *SciPy 1.13.1*, *StatsModels0.14.4* for scientific calculations, *Matplotlib 3.8.0*, and *Seaborn0.13.2* for plotting.

Network analysis was performed using a combination of *Gephi 0.10.1* software and *Python 3.6*. The relevant edge files in *csv* (comma-separated values) file format were generated in *Python*, imported in Gephi, and subjected to specific metric computations and network visualizations. From *Gephi*, the nodes table with all the computed parameters was exported in *csv* format and further filtered in *Google Sheets*. Also from *Gephi*, all the network renderings were exported as *pdf* files.

Home addresses were geocoded using the *Google Maps API* from *Python*, resulting in latitude and longitude coordinates for each entry. The geospatial visualization was made using *Google MyMaps*, which fed in data generated in the *Python* programs. County limits were set according to the *geojson* file offered by the community at geo-spatial.org, which is an extension of the Open Source Geospatial Foundation. This community generated the open source file format by merging and converting the official ANCPI (National Agency for Cadastre and Land Registration) files under the “Open Government License v1.0”.

### 2.2. Methods

#### 2.2.1. Study Design

This retrospective cohort study analyzed data for 131 months (January 2013–November 2023).

#### 2.2.2. Data Analysis

Age was calculated from the birthdate on file and extracted as full years of life. The length of stay was computed from the entry and exit dates, and was expressed in terms of a full hospitalization day. Descriptive statistics were employed to characterize the patient population and assess data distribution. Additionally, complex network analysis explored potential interactions and relationships between status asthmaticus and various comorbidities in selected populations. For the geospatial analysis, home location altitudes were generated from Google Maps data.

#### 2.2.3. Ethical Approval

The study received ethical approval from the appropriate ethics committees, numbers 38/24.11.2023 (university) and 10535/13.11.2023 (hospital).

## 3. Results

This study aimed to investigate the temporal patterns and prevalence of asthma diagnoses over 11 years (2013–2023) in a cohort of patients admitted to Victor Babes University Hospital in Timisoara. Due to the manner of collection, the data had a severity bias, as they represent primary respiratory care and do not fully reflect simple, uncomplicated cases that are dispatched to secondary or even tertiary care facilities. We acknowledge this bias and will endeavor to analyze the data accordingly.

### 3.1. Statistical Descriptions

The statistical parameters characterizing the length of stay and age of the selected lot are presented in Table 2. The data were split into patients presenting the J46 code (status asthmaticus) and those with just the J45 codes (asthma). All numerical values were rounded up to third decimals if they represented a floating-point value.

The corresponding kernel density plots were generated to visualize the distribution of these continuous variables, limiting the *x*-axis to the minimum and maximum values within the dataset (Figure 1).

There is a significant outlier group in the J46 sample that has 0 hospitalization days, which is highly improbable for a medical emergency. Therefore, the analysis will strive to take this into account.

### 3.2. Time Series

From a dynamic time perspective, simple counting was employed to track yearly evolution due to the relatively small number of cases in the J46 lot (Figure 2). As stated previously, the figure is rendered for the full-status asthmaticus sample as well as for the one that removed the outliers with 0 hospitalization days.

To analyze monthly trends (Figure 3), the number of cases per month was summed throughout the years and expressed as a percentage deviation from the total case number for typical asthma (J45 codes with all their sub-codes) and status asthmaticus (J46 code). The data were further split into males and females, stacking the columns to highlight monthly variations. Each male and female total value equaled 100%. Expressing this as a percentage allows a direct comparison between J45 and J46 evolution.

Females represented 58.36% of all J45 cases and 66.26% of all J46 cases, respectively, while males comprised the remaining 41.64% and 33.74%. Considering only patients with at least one full hospitalization day from the J46 sample, the proportions were further skewed towards females, with 70.31% cases, leaving just 29.69% males.

### 3.3. Geospatial Distribution

Using the patient’s home addresses, the geospatial distribution for asthma (J45 codes) and status asthmaticus (J46 code) is presented in Figure 4. The dots are plotted large enough to offer anonymity and still manage to convey the location. Certain geographical features are highlighted on the map. Only patients with home residences from Timis County were evaluated.

### 3.4. Comorbidities

An approach based on complex network (CN) analysis was employed to investigate the interactions among status asthmaticus and comorbidities. Specifically, each patient’s discharge data, expressed as International Classification of Diseases (ICD) version 10 codes, were transformed into lists, for example {J46, I10, J18.9}. All the list members became nodes, fully connected to create a patient’s edge list (each node was connected to all the others). Edges between nodes were assigned a weight of 1 to indicate identical contributions to the patient’s comorbidity profile. Returning to the previous example, the patient would have the following edges and weights: {(start = J46, end = I10, weight = 1), (start = J46, end = J18.9, weight = 1), (start = I10, end = J18.9, weight = 1)}. Each edge has a start and an end node, but we decided to use an undirected graph, meaning that it does not matter which is the start node and which is the end node of an edge since we do not know the exact disease causality and would like to model that accordingly, avoiding bias. Combining the edge lists for all the patients via addition meant that the more common edges would have higher weights, e.g., edge (start = J46, end = I10, weight = 10) was found in 10 patients. The final edge list was used to create the graph for our analysis.

This method does not exclude the more conventional co-occurrence data, which are still present in the form of node degree values (e.g., how many edges a node has) and edge lists, but the mathematical properties of the CN can extend the data insights. For example, as with status asthmaticus, it is interesting to discover the nodes that influence the network the most. This is reflected in the CN-specific measurement called betweenness centrality, which was therefore computed for each node. Moreover, in order to discover potential phenotypes, we employed the notion of communities. In CN, nodes that have stronger bonds with each other and lighter bonds with other nodes are bundled together in a community, much like a phenotype which presents a set of characteristics that are more common together. The CN properties tables are presented in the Appendix A for further reference.

From a visual perspective, all the CNs were rendered using the Force Atlas 2 layout to reveal communities more readily. Nodes were assigned colors corresponding to their respective communities, and their sizes were proportional with betweenness centrality, scaled logarithmically between 1 and 100. This scaling reflects the skewed data distribution by showing differences in nodes with a smaller value more clearly because the size increases more rapidly in relation to the betweenness centrality. Edges were rescaled from 0.1 to 1 in a linear fashion, proportional to their weight, in order to unclutter the render. The node’s label size is proportional to the node diameter. Any deviations from these parameters will be highlighted as needed. For the individual communities, all renders maximally filled the viewport. The latter’s size was constant for all images.

Comparing the status asthmaticus CN to the more common forms of asthma (J45 codes), the diffuse nature present in asthma is much more refined in the former exacerbation (Figure 5). While the graph density for status asthmaticus (0.028) is lower than that of common asthma (0.106), the significantly larger number of nodes and edges in the latter (2231 nodes, 70,708 edges vs. 174 nodes, 1595 edges) obscures potential phenotypic distinctions.

To illustrate the effect of alternative node sizing metrics, Figure 5c,d depict the same network as in Figure 5b, but node sizes are scaled by degree and weighted degree, respectively.

To characterize the phenotypic profile of each identified community, individual network visualizations were created, as presented in Figure 6, Figure 7, Figure 8, Figure 9, Figure 10 and Figure 11. The figures are arranged in descending order of community size within the network.

The main community is colored purple and has a 34.48% proportion of the nodes in the network. Status asthmaticus (J46) belongs to this community.

The main codes found in this cluster belong to the following disorders: status asthmaticus (J46), acute respiratory failure (J96.0), unspecified pneumonia (J18.9), chronic rhinitis (J31.0), unspecified asthma (J45.9), obesity (E66.0), hepatic steatosis (K76.0), hyperglycemia (R73), congestive heart failure(I50.0), hypertensive cardiomyopathy with congestive heart failure (I11.0), primary arterial hypertension (I10), chronic ischemic heart disease (I25.9), mitral valve insufficiency (I34.0), candidal stomatitis (B37.0), and urinary tract infection (N39.0).

The next cluster (Figure 7), depicted in green, covers 20.69% of the network and is skewed towards geriatric patients.

The main codes found in this cluster belong to the following illnesses: chronic respiratory failure (J96.1), bronchiectasis (J47), unspecified emphysema (J43.9), chronic obstructive pulmonary diseases with acute exacerbation/with acute infections (J44.0 + J44.1), unspecified interstitial lung disease (J84.9) sequelae of tuberculosis (B90.9), other specified sepsis (A41.8), spondylitis (M54.4), nutritional anemia (D53.9), loss of hearing (H91.9), obesity (E66.9), and hypercholesterolemia (E78.0).

The tertiary cluster, depicted in blue, covers 18.97% of the network and reflects poorly controlled diabetes mellitus.

The main codes found in this cluster belong to the following complaints: acute bacterial pneumonia (J15.8), Pseudomonas (B96.88), varicose veins of lower extremities with inflammation (I83.1), poorly controlled diabetes mellitus (E11.65), complicated diabetes mellitus (E11.8), chronic cholecystitis (K81.1), anemia (D50.8), and ischaemic cardiomyopathy (I25.5).

The quaternary orange cluster, representing 14.37% of the main network, is centered around infectious processes (Figure 9).

The main codes found in this cluster belong to the following ailments: unspecified chronic renal failure (N18.90), hyper transaminases (R74.0), unspecified heart failure (I50.9), atrial fibrillation and flutter (I48), hyponatremia hyperosmolar (E87.1), oxygen therapy (Z99.1), screening for viral diseases (Z11.5), asymptomatic hyperuricemia (E79.0), noncomplicated diabetes mellitus (E11.9), thoracic spondylosis (M47.84), hypokalemia (E87.6), and Parkinson diseases (G20).

The second-to-last cluster is small, representing just 9.2% of the network, and highlights the smoking phenotype.

The main codes found in this cluster belong to the following conditions: unspecified chronic obstructive pulmonary diseases (J44.9), pulmonary fibrosis (J84.1), smokers (Z72.0), unspecified chronic gastritis (K29.5), secondary pulmonary hypertension (I27.2), and unstable angina (I20.0).

The last cluster is minuscule, representing only 2.3% of the network, and seems to be a singular case, as proven by the fully connected graph with edge weights equal to 1.

The pink cluster codes are unspecified dementia (F03), anemia (D64.8), loss of hearing (H91.8), and unspecified thyrotoxicosis (E05.9).

## 4. Discussion

### 4.1. Statistical Description

#### 4.1.1. Data Evaluation

Despite the significant disparity in lot size (13,612 in J45 vs. 83 in J46), the age distributions were remarkably similar and close to normal distributions (kurtosis close to zero). Both groups had comparable standard deviations, indicating a similar variability in age distributions, but the J45 group has a more extended age range, which might be attributed to the larger sample size. However, this age discrepancy might also indicate a potential impact of status asthmaticus on disability-adjusted life years, which is consistent with findings linking acute asthma requiring mechanical ventilation to long-term mortality [17]. This latter hypothesis is supported by the fact that the code for mechanical ventilation (Z99.1) was present in 4 (4.82%) cases in the J46 lot (Appendix A—Appendix A) and 124 cases (0.91%) in the J45 lot. Running a chi-squared test for these data, we obtained a p-value of 0.0018, much smaller than the commonly used significance level of 0.05, leading us to conclude that the proportion of patients needing mechanical ventilation is not due to chance.

These findings correlate with the length of stay statistics (Table 1, Figure 1b). The disease burden was higher for the J46 patients, with a significantly longer average length of stay (6.024 vs. 4.48 days) and maximum values (20 vs. 7 days). Length of stay variability for J46 patients was slightly lower than that of J45 patients, as indicated by standard deviation. The J45 group had a right-skewed, heavy-tailed distribution (skewness > 0, kurtosis > 3), specific since this is an emergency hospital—only high-risk patients are admitted for longer stays—while the J46 sample was close to a normal distribution, with a slightly longer left-side tail. It is worth exploring the fact that even though J46 is a medical emergency, 19 people had 0 days spent in the hospital, meaning that they were discharged on the same day. Investigating the outlier sub-dataset further, their stay ranged from 4 to 19 min, with a mean of 11.47 min, with six patients arriving during the night on-call shift. They did not have any other diagnoses except J46 on file, with only one person being marked as a social case. This indicates that these patients were ambulance arrivals, rerouted to other specialty emergency hospitals, more than likely because their symptoms were not consistent with a respiratory emergency. This lot was further excluded from our analysis since they do not represent true J46 cases.

Regarding gender, females were majoritarian in both the J45 (58.36%) and J46 (70.31%) groups. The odds ratio for developing status asthmaticus in women compared to men was 1.401 (95% CI: 0.888–2.212). Although this suggests a higher likelihood of status asthmaticus in women, because the confidence interval includes 1, the difference might not be statistically significant. Further prospective research is needed to explore this gender disparity.

#### 4.1.2. Clinical Implications

Patients with status asthmaticus (J46) experience a significantly higher disease burden compared to those with asthma (J45), as evidenced by the longer average length of stay (6–8 days) than the average length of stay for hospitalized asthmatics (3.2 days) and more severe clinical presentations [19].

The data suggest a potential gender disparity in the development of status asthmaticus, similar to the literature outcomes: women have a higher prevalence of asthma. They are more likely to have severe asthma compared to men due to hormonal factors, anatomical particularities, and environmental exposure [20].

Patients with status asthmaticus are more likely to require mechanical ventilation, with a higher rate of mortality (10–25%, compared to typical asthma at 1–5%), suggesting a more severe course of the disease. This finding aligns with previous research linking acute asthma requiring mechanical ventilation to long-term mortality [21,22].

### 4.2. Time and Space Distributions

#### 4.2.1. Data Evaluation

Time and space distributions should be correlated, as they represent facets of the same story. Comparing Figure 4b–c, the difference is striking: J45 spans the whole county, while J46 effectively spans only the left (western) half. Since the time collection period is significant (11 years), this spatial disparity suggests a non-random distribution despite J46’s lower prevalence. Further exploring the actual terrain geography gives us clues regarding this distribution. The J46 points are located in the plain, while J45 spans across multiple landforms, with a plain predominance. From an altitude perspective, almost all J46 cases are in the lower Banat plains, at altitudes less than 100 m. The outlier from this plain paradigm is the one on the far right, which is actually located in the Timis river meadow, in one of the lowest places in that area. It is known that asthma, particularly acute asthma, is sensitive to particulate matter (PM), especially in women. The literature shows that PM tends to be trapped in a shallow, stable surface layer, decreasing with an increase in altitude and/or horizontal distance from the main road [23,24]. This is consistent with our findings, where asthma cases follow the main roads and lower altitudes. The other studies found that correlated geospatial data with asthma intensity were carried out in pediatric patients in a mountainous setting, therefore lacking correlation to our approach [25].

To further explore this PM hypothesis, we looked for correlations with pollution data. However, there are a lot of controversies about data quality and accuracy [26,27], so we decided to use just the official European Environmental Agency data, which has the most credibility (Table 3) [28]. The statistical correlations between pollution data and yearly variations are significant for asthma (J45) and particulate matter. From a statistical standpoint, J46 and J45 yearly variations are not similar, and neither of them follow a linear progression. Both of them showed lows during the COVID-19 years.

The lack of the expected correlation between J46 and PM levels (PM2.5 and PM10) has several explanations, the first being that the data collection method for PM involves mainly city networks, disregarding the countryside. The main city is Timisoara, home to 33.077% of the county’s population. J45 had a majority (75.28%) of its subjects inside the city limits (Figure 4a), which varied from the sensor data. In comparison, J46 had only 25 cases (39%) inside the Timisoara city limits. Although this is proportional to the population density, the lack of data outside city limits to fit this granularity makes us unable to make a comparison. Another justification is that at least one unaccounted variable influences J46’s connection to particulates; socio-economical, incorporating factors like access to medical care, or environmental. This can be due to a seasonal spike in allergens, either pollens as recorded in the higher levels of May and June or the endemic Arthemisia in August (Figure 5c). Cold weather definitely plays a role, as proven by the levels in those months for both J45 and J46 (Figure 5a,b).

Another reason might lie in the way individual comorbidities interact, not only with the environment but with themselves. For this, let us refer to the complex networks from Figure 5, Figure 6, Figure 7, Figure 8, Figure 9, Figure 10 and Figure 11 and the discussion in Section 4.3.

#### 4.2.2. Clinical Implications

Environmental Factors and Asthma:

The spatial distribution of asthma cases, particularly status asthmaticus (J46), is significantly influenced by environmental factors, primarily exposure to particulate matter (PM). This highlights the importance of considering environmental determinants in the development and management of asthma.

Geographic Disparities: The concentration of asthma cases in specific geographic regions, such as the lower Banat plains, suggests that environmental factors in these areas may contribute to a higher prevalence of asthma.

Air Quality and Asthma: Exposure to air pollution, particularly PM, is associated with increased asthma risk and severity. This finding underscores the need for improved air quality monitoring and interventions to protect public health.

Individual Variability: The impact of environmental factors on asthma may vary among individuals due to differences in genetic susceptibility, comorbidities, and socio-economic factors. Further research is needed to better understand these individual variations and tailor prevention and management strategies accordingly.

Results relevance: Timiș County, Romania’s largest county, is characterized by a medium-high population density, with a significant portion concentrated in the metropolitan area of Timișoara, the country’s third-largest city (Figure 4a). This urban–rural mix and the county’s plain-prevalent environmental factors, including exposure to pollutants, contribute to unique patterns in asthma prevalence and severity.

Data Limitations: The quality and availability of pollution data can pose challenges in assessing the relationship between air pollution and asthma. This highlights the need for improved data collection and analysis methods to enhance our understanding of these factors.

### 4.3. Commorbidies Clusters

#### 4.3.1. Data Evaluation

A discussion on the limitations inherent to this retrospective study is now in order: there is a known rater variability in diagnosis coding. Physicians and registrars might inadvertently alter some nuances due to time pressure or the national requirement for each discharge sheet to meet minimum Diagnostic Related Group (DRG) criteria. This can introduce a lot of noise, and here is where complex network metrics can help underscore the truth. Representations based on node degree and weighted degree (Figure 5c,d) are informative in certain contexts, but in a dense network such as the ones for status asthmaticus or asthma, they fail to discriminate between nodes effectively. Looking at the clusters, most nodes seem to have equal importance, but when switching to a centrality-based metric (Figure 5a,b), the image starts to highlight the structural positions of a node, like bridges/connectors, reducing the noise (spurious connections).

The clusters were defined by considering not only the number of edges but also their weight, i.e., the power that the edge has. The resulting six clusters are presented quantitatively in Table 4, in descending order based on node numbers.

The largest cluster is, as expected, connected to all the other clusters; however, it stands out in the connection it has to the blue cluster. This suggests that elements from both of them tend to appear in the same patient, more so than any other inter-cluster connection. Also, the purple–orange interconnection is very strong relative to node size. In a broad interpretation, the purple cluster is the “glue” that holds together the network, indicating a common denominator for the patients.

The green cluster is more isolated than the purple one, with which it shares the strongest connection. The remaining four clusters are very isolated, indicating clearly defined patient presence, with very few overlaps.

#### 4.3.2. Clinical Implications

In order to better conceptualize the clusters, we imagined them as typical cases that present the underlying disorders in that community alone and recreated the clinical presentation that describes them.

The central community in Figure 6 describes a patient who is admitted with a clinically severe altered state, acute respiratory failure signs (J96.0) due to severe airflow limitation, and with abuse of/or inadequate inhaler technique for corticosteroid/long-acting beta two agonists devices (ICS/LABA). This severe exacerbation, life-threatening, seems to appear in an infectious context (pneumonia J18.9) in a patient with comorbidities that make him susceptible to poor outcomes. The presence of multiple co-existing conditions like metabolic syndrome (obesity—E66.0, hyperglycemia R73, primary arterial hypertension I10) or heart disease with long-term cardiovascular deficiency (I50.0, I11.0, I25) can have a cumulative effect, and often worsen consequences of asthma exacerbations. Iatrogenic oral candidosis (B37.0) is, as expected, very often found secondary, as mentioned, to inadequate inhaler techniques or abuse of ICS, which can increase the risk of fungal infections due to immunosuppressive effects.

Chronic inflammation in the upper airway (rhinitis J31.0) suggests an atopic status, too, and can contribute to airway hyper-responsiveness and increase the risk of asthma exacerbations.

Because lung inflammation can lead to asthma, airway dysfunction related to pneumonia can cause a severe attack. Asthma is associated with an increased risk of pneumonia, but there is limited information on the underlying mechanism of this predisposition. As bacteria or viruses replicate inside the lung tissue, the body’s natural immune defenses flood the lungs with mucus, making the scenario worse [29]. Treatment-related immunomodulation, such as inhaled corticosteroids in asthma (a leading treatment option for asthma) or disease-related inflammation within the epithelial milieu, could modify the propensity and density of microparticle carriage (pathogens and allergens) [30]. ICS can also alter the composition of the airway microbiome, which may influence the density of microbial carriage. Together, they might increase the risk of developing pneumonia or other respiratory infections.

Obesity is a significant risk factor for asthma morbidity. Obesity, together with high blood pressure, insulin resistance, and abnormal cholesterol levels, is associated with metabolic syndrome. This cluster has the potential to cause long-term, low-grade inflammation, which exacerbates asthmatic airway inflammation and raises the risk of severe exacerbations and difficulty controlling symptoms. Morbid obesity can worsen asthma symptoms and increase the risk of asthma exacerbations due to mechanical effects on the lungs, further compromising oxygen exchange during an asthma attack [31,32,33].

Decreased oxygen delivery: Heart disease can make it more difficult for the heart to pump blood efficiently. As a result, the body’s tissues—including the lungs—receive less oxygen. Heart problems can make things much worse during a severe asthma exacerbation, when oxygen demand is high and airways are restricted [34]. 

Elevated cardiac strain: More significant effort in breathing associated with severe asthma puts additional strain on the heart. Heart failure may result from the heart’s inability to withstand this extra strain due to pre-existing cardiac disease [35].

According to a recent meta-analysis [36], *bronchiectasis, allergic rhinitis, nasal congestion, allergic conjunctivitis, hypertensive cardiomyopathy, COPD, and other chronic respiratory diseases are strongly associated with severe asthma*.

To gain a clearer insight into population clusters, we aimed to outline some key findings regarding the cohort. Various host factors, including genetic predispositions and concurrent health conditions, along with environmental influences such as viral infections, cigarette smoke, and indoor/outdoor air pollution, play a role in the development of steroid-refractory T2-high asthma [37]. Thus, this core studied population group reflects ***inadequate disease control over time***, ***probably**at Th2-high***, with ***notable symptoms*** and ***systemic and local inflammatory*** status as a burden, reflecting a ***vast range of comorbidities***.

The cluster in Figure 7 seems to describe older patients (spondylitis, M54.4; loss of hearing, H91.9) with a prolonged duration of asthma disease (they arrived with pre-existing chronic respiratory failure status, J96.1, and the presence of the bronchiectasis, J47), less severe atopic features, higher probability of being smokers or former smokers (presence of emphysema J43.9), and with forced expiratory volume in 1 s (FEV1) and less reversibility, which are similar features to those of COPD (J44.0/J44.1).

While not directly related to asthma, some interstitial lung diseases (like J84.9, also present in this cluster) can mimic asthma symptoms and may be challenging to differentiate from status asthmaticus [38]. Scarring of the parenchyma, bronchiectasis, and bronchial stenosis from past tuberculosis infection (B90.9 presence) can increase the susceptibility to asthma and worsen asthma symptoms through airflow limitation [39]. The coexistence of bronchiectasis in asthma patients can complicate the clinical picture and is responsible for worsening symptoms associated with frequent exacerbations and higher disease severity [40]. Severe extended asthma seems to be associated with a high rate of bronchiectasis [36,41].

This cluster reflects a progressive asthma disease, probably Th2-low, at an older age with less noticeable symptoms and less significant comorbidities.

The blue tertiary cluster (Figure 8) centers on the main comorbidity—poorly controlled diabetes mellitus (E11.8)—and is the perfect substrate for aggressive lung parenchyma infections (J15.8) caused by Pseudomonas aeruginosa (B96.88).

Patients with poorly controlled diabetes are at an increased risk of infections caused by opportunistic pathogens, including *Pseudomonas aeruginosa*. This bacterium is known for its ability to cause severe lung infections, particularly in immunocompromised individuals or those with underlying respiratory diseases. The combination of poorly controlled diabetes and *Pseudomonas* infections can lead to a vicious cycle. The infection exacerbates asthma symptoms, which can lead to a further deterioration of glycemic control due to stress and inflammation. This cycle can increase hospitalizations, prolong recovery periods, and contribute to a higher mortality risk [42].

Diabetes mellitus (E11.8) is one the most substantial risk factors for cardiovascular disease and, in particular, for ischemic heart disease (I25.5) due to several underlying mechanisms: coronary stenosis, coronary microvascular disease, and dysfunctional ion channels [43]. I25.5 also has essential connotations in our cluster and represents an important relay in the above complex network, once again emphasizing the metabolic imbalance of diabetes.

This cluster of status asthmaticus is centered around ***poorly controlled diabetes mellitus***, highlighting ***a critical intersection of respiratory and metabolic health***.

In the orange cluster (Figure 9), the interplay between asthma and these conditions emphasizes the need for comprehensive management strategies addressing respiratory and systemic health to improve patient outcomes. In this complex network of interactions between various diseases associated with the status asthmaticus, attention is drawn by multiple organ insufficiency (chronic renal failure, N18.90; unspecified heart failure, I50.9; respiratory failure with need for oxygen therapy, Z99.1). The clinical picture suggests a severe acute episode superimposed on a pre-existing chronic pathology (heart, renal (N18.90), metabolic, +/−neurological/neurodegenerative diseases—G20). Viral infections, such as influenza or COVID-19 (Z11.5), can trigger a systemic inflammatory response, leading to various complications. These may include cardiac arrhythmias, such as atrial fibrillation or flutter (I48), hepatic dysfunction (elevated liver enzymes, R74.0), and electrolyte imbalances (hyponatremia E87.1, hypokalemia E87.6), often secondary to poor glycemic control in diabetes or as a consequence of cardiac and asthma medications. Additionally, viral infections can exacerbate acute severe asthma, potentially leading to respiratory failure.

Viral infections can significantly impact a person’s health, mainly when we refer to viruses like influenza or COVID-19, and their effects can be even more pronounced in individuals with pre-existing conditions like asthma, heart failure, and diabetes mellitus. Atrial fibrillation or flutter may be a generalized response to inflammation of severe viral illness [44] or an abuse of inhaled short-acting B2 agonists (SABA), which can produce direct arrhythmia or secondary hypokalemia [45]. Liver involvement in acute respiratory viral infection is well known [46] and could reflect disease severity [47].

This cluster expresses the wide-ranging impact that a ***potentially severe viral respiratory infection*** can have in a patient with ***multiple chronic pathologies and asthma***. ***The T2-high subgroup*** appears to be particularly exacerbation-prone to viral infections [48].

The teal cluster (Figure 10) is the prototype of longstanding severe chronic fix asthma obstruction (J44.9 due to asthma, COPD, or asthma/COPD overlap) and is very well linked with the smoker’s node (Z72.0). Pulmonary hypertension (I27.2) and *cor pulmonale* are complications of severe asthma/COPD as a consequence of pulmonary hypoxic vasoconstriction [49]. Unstable angina (I20.0) indicates significant cardiovascular compromise, which can be exacerbated by the hypoxia and increased respiratory effort associated with status asthmaticus. While smoking is an indubitable risk factor for fixed obstruction asthma/COPD, it can also increase the risk of developing pulmonary fibrosis (J84.1). Also, bronchial asthma, airway remodeling, and lung fibrosis could be successive steps in one process.

A subgroup of patients with long-term persistent asthma may develop irreversible or fixed airflow obstruction (FAO) [50], regularly refractory to long-term pharmacologic treatment [51]. It is hard in clinical practice to distinguish between asthma with FAO and COPD or a case of them overlapping. The observed progression of some cases of persistent asthma towards FAO finds support in epidemiological studies. These studies show a high prevalence of co-existing asthma and COPD diagnoses, suggesting a potential link. Additionally, patients with active, persistent asthma are significantly more likely to receive a COPD diagnosis later in life. The prevalence of asthma patients with FAO increased with age and in ex- and current smokers.

Among the most commonly proposed risk factors are blood and sputum eosinophilia, adult-onset asthma, older age, male sex, and a history of smoking [52]. Asthma–COPD overlap syndrome (ACOS) defines those patients with a substantial smoking history and consequent airflow obstruction but who also have overlapping features of asthma (bronchodilator reversibility, eosinophilia, atopy) [32].

In the pathophysiology of severe and chronic asthma, lung fibrosis is frequently observed and is linked to both the severity of the disease and treatment resistance. Therefore, it can be considered an irreversible effect of airway remodeling and inflammation brought on by asthma [53].

We will title this cluster of longstanding severe ***chronic fix asthma obstruction*** on heavy or former ***smokers*** the ***“Asthma smoking phenotype”***.

The pink cluster (Figure 11) is an outlier, consisting largely of a single patient, and therefore lacking phenotypic significance.

In other words, with the help of engineering techniques such as a complex network, we tried to identify the characteristics of the different subgroups of patients with status asthmatichus in Timis County to outline regional clinical phenotypes. Data limitations: lack of lung function (like spirometry) and biomarker data may limit phenotyping accuracy. The identified phenotypes may not fully reflect the underlying pathogenic mechanisms but only suggest them, correlating the information obtained with that reported in the literature.

The proposed clinical phenotypes of status asthmaticus are presented in Table 5, with proportions representing the number of nodes in the cluster relative to the entire network. The endotypes and phenotypes are assumed to be as such since the retrospective data lack the variables to confirm it, but the clinical tableau suggests it.

The correlation with Table 1 is not immediately apparent for the purple and orange clusters (presumed T2-high endotypes). These specific clusters have a definite cytokine aspect, reflecting years of inflammation. In Table 1, the atopic allergy cluster refers mainly to the pediatric population, which is absent in our case due to the specific data collection criteria. Nothing in our data highlights the eosinophilic phenotype, so we propose these two clusters as specific ones for our region. A very important idea about these clusters is that both could have much better outcomes with proper prevention techniques.

The other three clusters have correspondents in the literature. The non-atopic, neutrophilic phenotype from Table 1 corresponds to our blue cluster, the persistent Mycoplasma pneumonia infections underlining that the specific aspect is in the poorly controlled diabetes diagnosis, which indicates a possible intermediation option. The neutrophilic asthma phenotype from Table 1 has a strong correlation with the green cluster in our research. The overlap endotype is present in both tables, and the specificity brought by our data relates to smoking, confirming the “Dutch hypothesis” and indicating a clear intervention point for our population.

## 5. Conclusions

In this retrospective study, we explored different facets of status asthmaticus, comparing it with asthma in a relatively homogenous population in Timis County over 11 years. The main insights answer the original research questions.

Question 1 was “Are there temporal patterns in status asthmaticus prevalence over the last 11 years?”, and it was proven that the disease burden in status asthmaticus is significantly higher than in asthma, as proven by the much longer hospital stays. There is a definite variability in annual and monthly prevalence, with no statistical connection between asthma and status asthmaticus variations. Certain periods exhibit higher incidence rates, suggesting a contribution from potential seasonal or environmental factors. However, further investigation is needed due to the retrospective nature and limited certifiable pollution data.

Question 2 was “Do demographic factors such as age and gender influence the risk of status asthmaticus?”, and it was determined that both gender and age influence the prevalence of status asthmaticus. As expected, females are disproportionately affected (70.31%). In asthma and status asthmaticus, older adults prevailed (~50% over 60 years), but status asthmaticus proved to lower maximum age by 8 years, probably due to the disease burden.

Research question 3 was “Are there discernible status asthmaticus associations with known triggers?”, and we provided unique insights specific to the Timis County area. Status asthmaticus cases were concentrated in low-altitude locations, reflecting environmental or other socio-economic factors (like low-income areas). On the other hand, asthma had a disproportionately high prevalence in the main city, Timisoara. The latter’s variance was also statistically correlated with particulate matter concentrations registered in the city. These results should be the reasons for targeted public health interventions in the highlighted areas because it is known that air pollution exposure (higher ozone, small PM, NO_2_) activates the inflammatory-specific cascade and triggers the crisis, even at levels well below European regulatory limit values. Moreover, given the diverse demographic and environmental characteristics of Timiș County, which is representative of Romania’s broader population and environmental conditions, the findings from this study may have national implications for understanding and managing asthma.

The last research question, “What are the interactions between status asthmaticus and comorbidities? How do they reflect existing literature phenotypes?”, is the one that provided the most surprising results. Using a methodological innovation in the field, like complex network analysis, the interactions between comorbidities revealed five proposed clinical phenotypic clusters associated with status asthmaticus. The identification of clusters linking asthma with other conditions (e.g., diabetes, smoking-related issues) provides valuable data for research into disease interactions and multimorbidity. Almost half (48.85%) of patients with status asthmaticus have higher cytokine-specific inflammatory phenotypes that can be successfully controlled using steroid approaches. The lack of proper control and opportunistic infections are responsible for their exacerbation, which is good news since these are all preventable, and it highlights the importance of focusing on prevention and management strategies of asthma symptomatology and treatment awareness. The presumed T2-low clusters reflect the aging population (20.69%) and poorly controlled diabetes (18.97%). While the first is a natural consequence, both these groups would have benefited dramatically from different lifestyle choices if implemented early enough. Therefore, age-adapted asthma management and the identification of poorly controlled diabetic asthma patients could lead to better-personalized asthma/diabetes care and outcomes. This leads us to conclude that aggressive screening and early interventions for these clinical phenotypes could generate different outcomes in a medium timeframe (25 years). The last phenotype is an overlap of asthma and COPD, but the smoking prevalence in this group indicates that the same lifestyle changes are crucial for this group as well.

In conclusion, even in the absence of lung function data (like spirometry), clinical phenotyping based on observed behavior can provide valuable information for more targeted healthcare planning and resource allocation for patients with severe asthma in Timiș County. This approach may help to optimize treatment and identify patients who may benefit from targeted therapies or closer monitoring. Furthermore, this innovative approach could be applied to other chronic diseases or geographical contexts.

Further prospective interdisciplinary research is needed to explore the causes of the observed patterns and to develop targeted clinical trials to validate our findings and assess the impact of phenotype-specific interventions. This study can also be a valuable reference for similar regions, either in methodology or by localized healthcare strategy transfer, while also having the potential to improve patient care pathways in our institution.

## Figures and Tables

**Figure 1 jcm-13-06615-f001:**
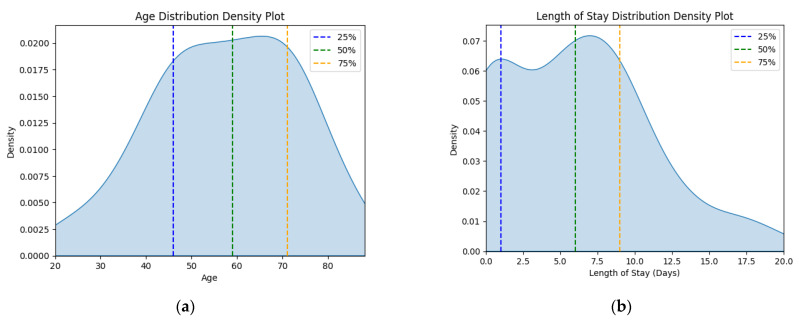
Density plots for (**a**) age variable and (**b**) length of stay variable for status asthmaticus.

**Figure 2 jcm-13-06615-f002:**
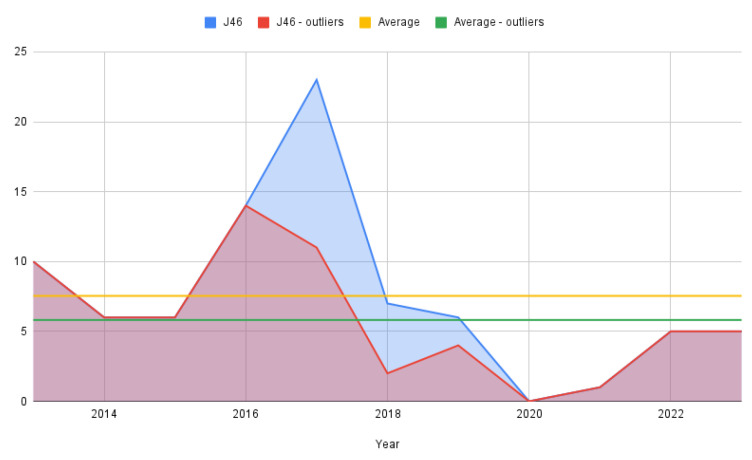
Status asthmaticus cases per year.

**Figure 3 jcm-13-06615-f003:**
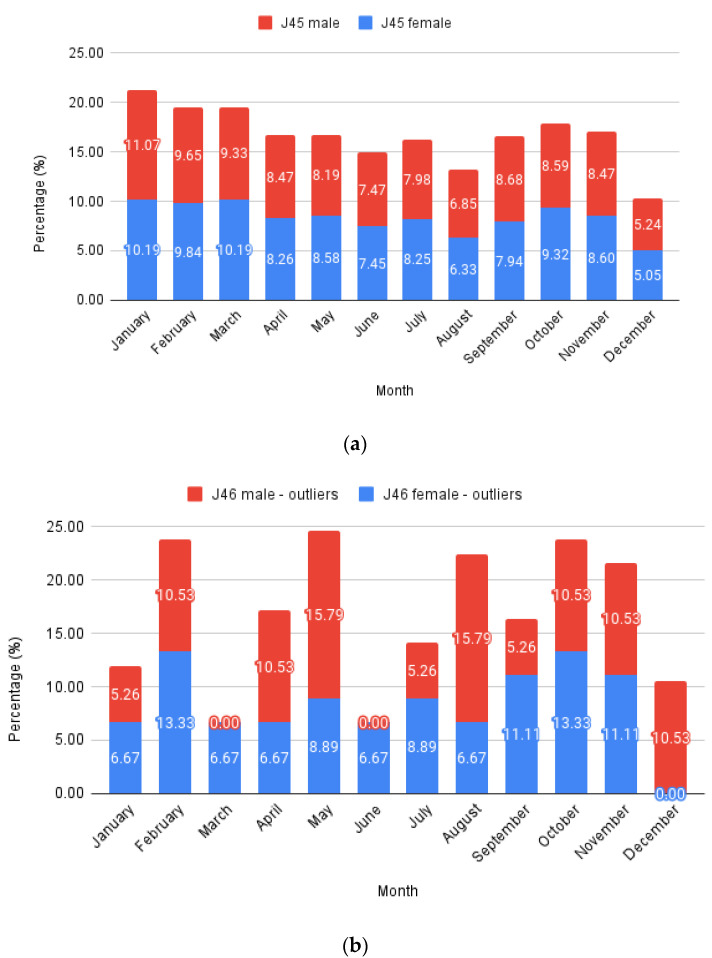
Monthly relative values: (**a**) J45; (**b**) J46—outliers.

**Figure 4 jcm-13-06615-f004:**
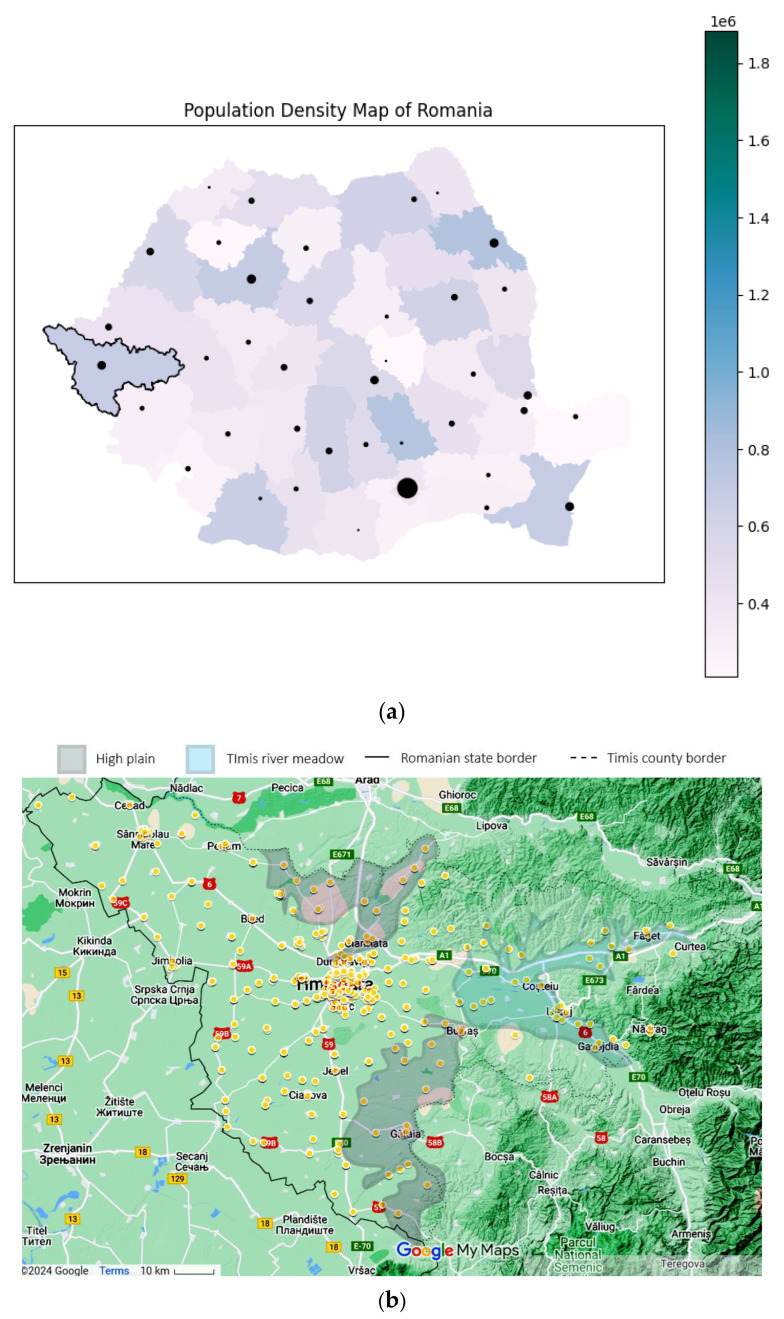
Geographical data, representing the location of the study in a national context: (**a**) Timis County, marked with a black contour (county color scaled to population density. Municipality size proportional to city population) and patients’ origins for (**b**) J45 asthma—yellow and (**c**) J46 status asthmaticus—red, with outliers as gray dots. Statistical data from the Romanian National Statistics Institute. Geospatial data are a correlation between Google Maps and [18].

**Figure 5 jcm-13-06615-f005:**
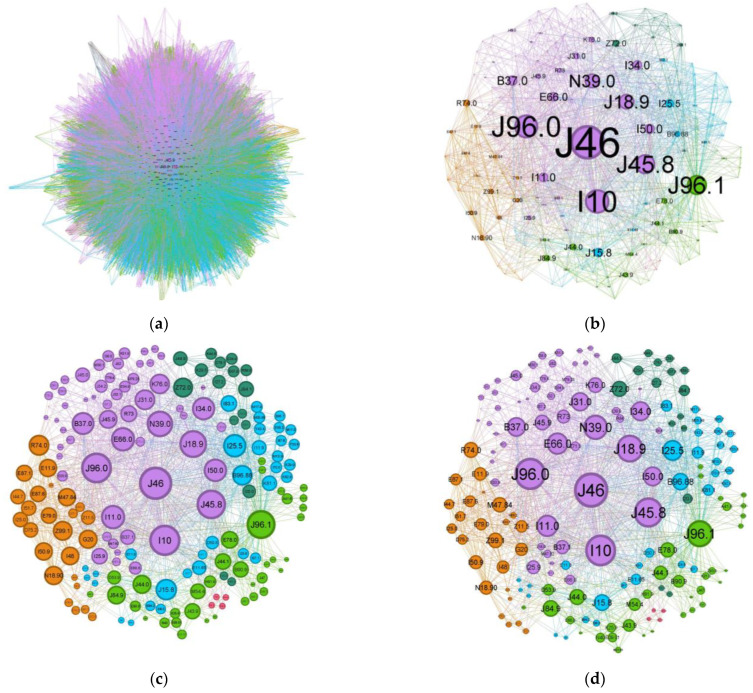
Comorbidities CN for (**a**) regular asthma (J45 codes) and (**b**–**d**) status asthmaticus (J46 code).

**Figure 6 jcm-13-06615-f006:**
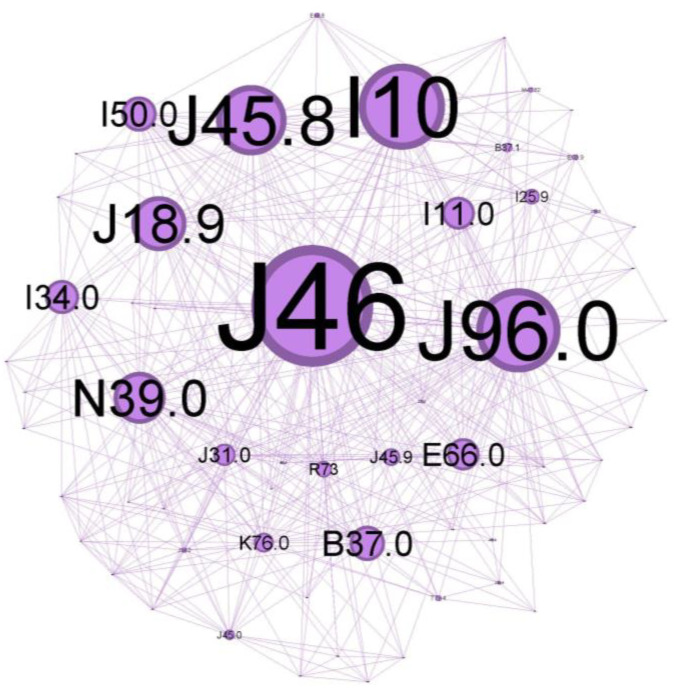
Purple cluster clinical phenotype: central community—*inadequate disease control over time*, with *noticeable symptoms* and *systemic and local inflammatory* status as a burden, reflecting a *vast range of comorbidities*.

**Figure 7 jcm-13-06615-f007:**
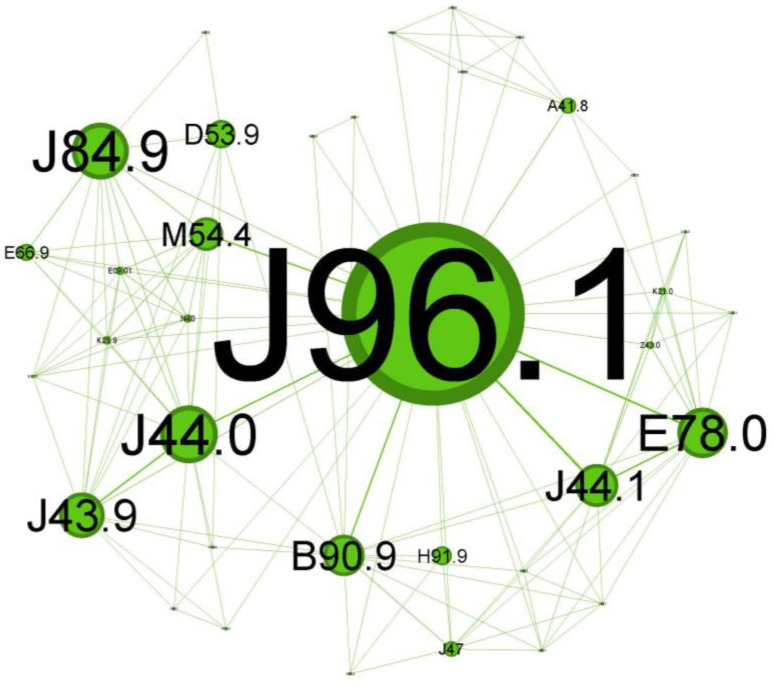
Green cluster clinical phenotype reflects *a progressive asthma disease at an older age* with not-so-noticeable *symptoms* and *less significant comorbidities*.

**Figure 8 jcm-13-06615-f008:**
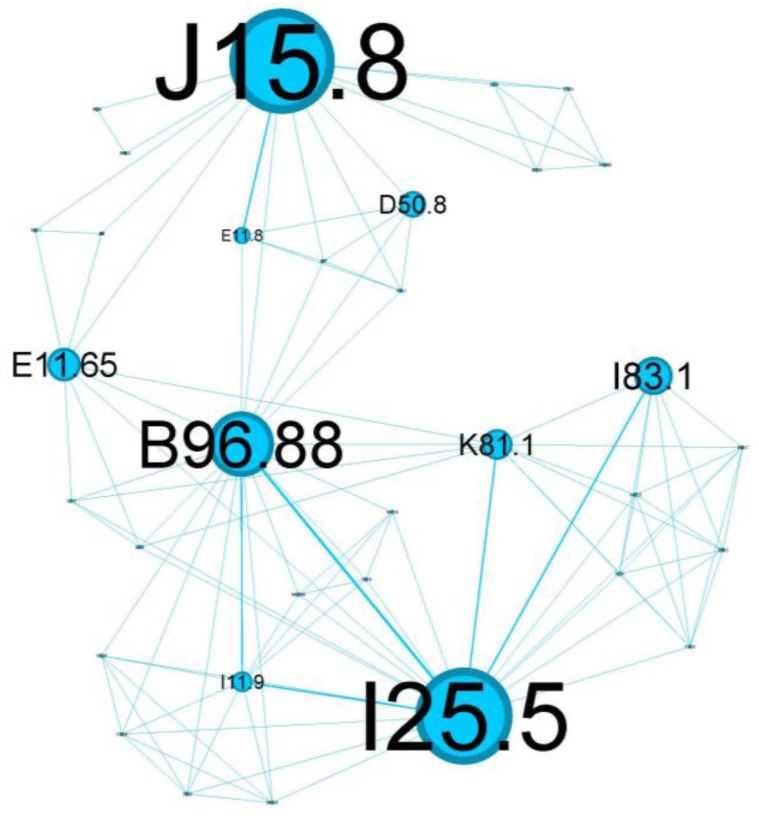
Blue cluster clinical phenotype is centered around *poorly controlled diabetes mellitus*, highlighting *a critical intersection of respiratory and metabolic health*.

**Figure 9 jcm-13-06615-f009:**
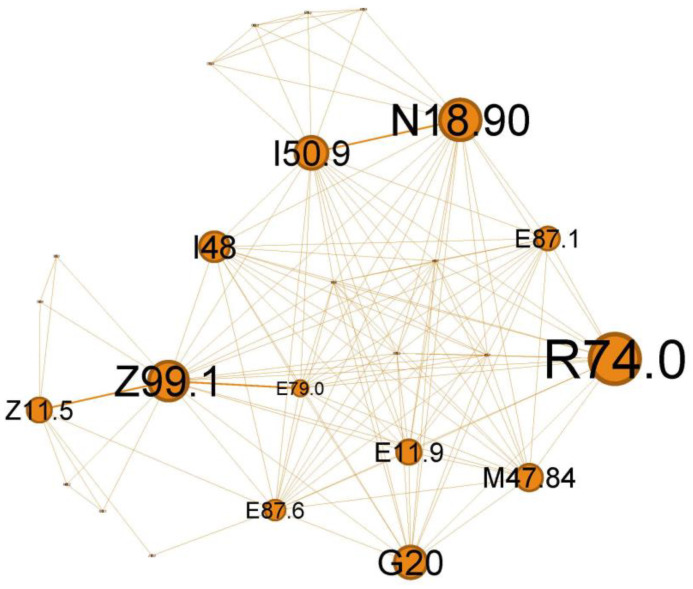
Orange cluster clinical phenotype, expressing the wide range of potential severe viral respiratory infections that can occur in a patient with *multiple chronic pathologies and asthma*.

**Figure 10 jcm-13-06615-f010:**
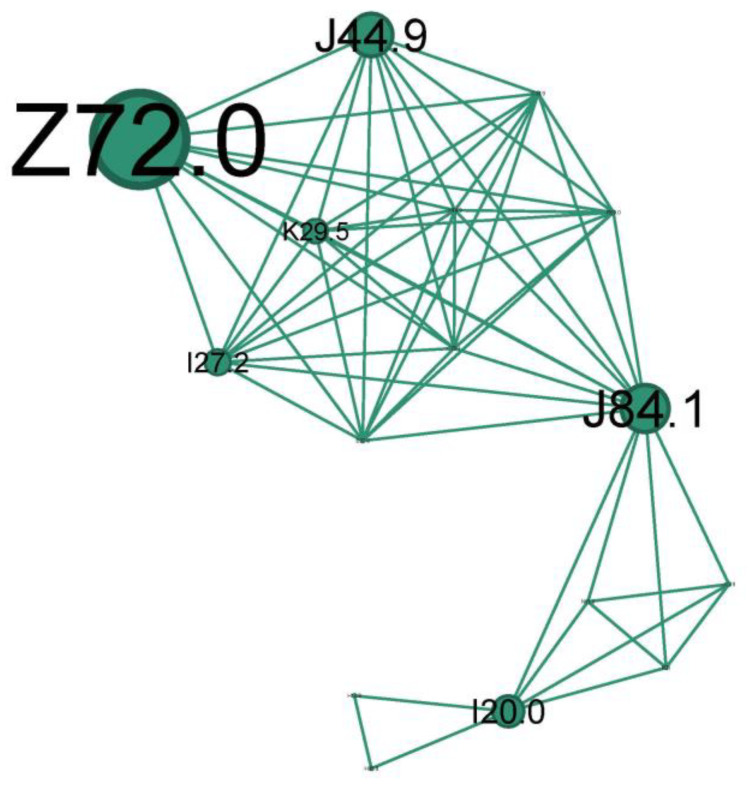
The teal cluster clinical phenotype—*asthma smoking phenotype. Longstanding severe chronic fix asthma obstruction* on heavy or former *smokers*.

**Figure 11 jcm-13-06615-f011:**
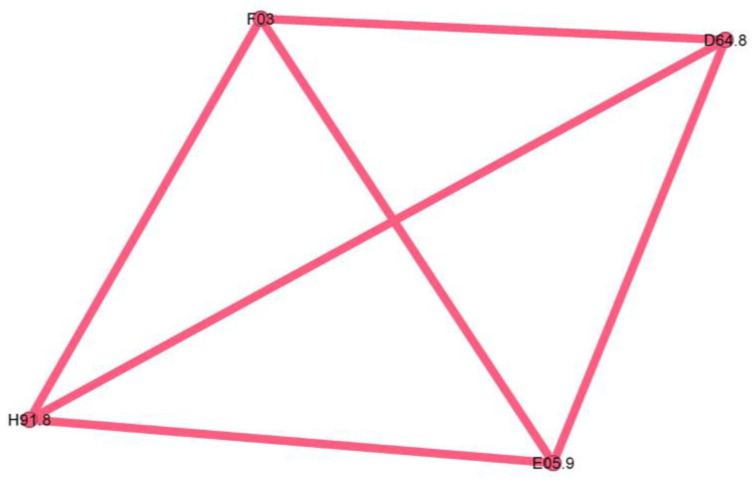
Pink cluster clinical phenotype—an outlier, consisting largely of a single patient, lacking phenotypic significance.

**Table 1 jcm-13-06615-t001:** Main endotypes and phenotypes (CRSwNP—chronic rhinosinusitis with nasal polyps; COPD—chronic obstructive pulmonary disease).

Endotype	Phenotypes	Age/Onset	Substrate/Comorbidities	Response to Treatment	Source
**T2-high**	Atopic allergic	Early	Increased serum-specific IgE	Good with corticosteroid	[11]
	Eosinophilic	Late-adult onset	Chronic rhinosinusitis with nasal polyps (CRSwNP)	Steroid-resistant	[12]
**T2-low**	Non-atopic neutrophilic	Adult	Obesity, smoking, and persistent infection with atypical bacteria (Mycoplasma pneumoniae and Chlamydophila pneumoniae)	Steroid-resistant	[11]
	Smokers	Older	Frequent exacerbation, lower lung function,	Steroid-resistant	[13]
	Obesity-associated	Middle-aged, women	Severe symptoms; average preserved function	Steroid-refractory	[13,14]
	Neutrophilic	Geriatric	Immunosenescence		[15]
**Unclear**	“Dutch hypothesis” or Overlap	Adult	Hypertension, elevated cholesterol/hyperlipidemia, arthritis, depression, obesity, gastroesophageal reflux disease		[16]

**Table 2 jcm-13-06615-t002:** Statistical descriptions of the dataset.

Measurement	Age		Length of Stay	
	J46	J45	J46	J45
Count	83	13612	83	13612
Mean	57.590	59.619	6.024	4.484
Standard deviation	15.957	14.813	4.983	5.941
Minimum value	20	18	0	0
First quartile (25%)	46	51	1	0
Median quartile (50%)	59	62	6	1
Third quartile (75%)	71	70	9	8
Maximum value	88	95	20	7
Skewness	−0.260	−0.567	−0.597	1.883
Kurtosis	−0.508	−0.044	−0.088	5.854

**Table 3 jcm-13-06615-t003:** Pollutant correlation matrix. The background color gradient indicates the strength of the correlation, ranging from red (low correlation) to yellow (moderate correlation) to green (high correlation).

	NO2	O3	PM2.5	PM10	J45	J46
**J45**	0.389	−0.051	0.576	0.684	1	0.116
**J46**	−0.061	−0.305	0.373	0.366	0.116	1

**Table 4 jcm-13-06615-t004:** Cluster sizes and number of inter-cluster connections.

No.	Cluster	Nodes	Edges	Connections to					
				Purple	Green	Blue	Orange	Teal	Pink
1	**Purple**	60	416	-	134	204	174	87	12
2	**Green**	36	138	134	-	49	27	20	4
3	**Blue**	33	108	204	49	-	15	5	0
4	**Orange**	25	133	174	27	15	-	5	0
5	**Teal**	16	58	87	20	5	5	-	0
6	**Pink**	4	6	12	4	0	0		-

**Table 5 jcm-13-06615-t005:** Timis County-specific status asthmaticus clusters.

Color	Presumed Endotype	Clinical Phenotypes	Substrate/Comorbidities	%	Recommended Attitude/Treatment
Purple	T2-high	High inflammatory status	Inadequate disease control, metabolic syndrome, heart disease with long-term cardiovascular deficiency	34.48%	Promptly treat asthma exacerbation cause; Check inhalator technique; Asses proper control of comorbidities; Asses and combat environmental influences; Step up asthma treatment; Asses asthma biomarkers and biological therapy.
Orange	T2-high	Infections	Oxygen therapy, pre-existing chronic pathology (cardiovascular, renal, neurological, and liver failure)	14.37%	Aggressively treat infections; Temporarily stepup of asthma treatment; Asses proper control of comorbidities; Asses asthma biomarkers.
Green	T2-low	Senescence	Interstitial lung disease, bronchiectasis, emphysema	20.69%	Check alternative diagnosis/treatment; Add BD; Check for chronic lung infections; Asses asthma biomarkers.
Blue	T2-low (maybe?)	Poorly controlled diabetes	Aggressive lung parenchyma infection cycle, ischemic heart disease	18.97%	Aggressively treat infections; Refer to/seek advice from a metabolic specialist.
Teal	Overlap	Asthma/COPD	Smoking, secondary pulmonary hypertension	9.2%	Smoking cessation; Treat like asthma.

## Data Availability

The datasets generated during and/or analyzed during the current study are not publicly available due to privacy (GDPR) restrictions. However, anonymized datasets may be available upon reasonable request and with appropriate ethical approval. Requests to access the datasets should be directed to Dr. Oancea (oancea@umft.ro).

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
