# Peer review of "A Multifaceted Exploration of Status Asthmaticus: A Retrospective Analysis in a Romanian Hospital"

_jcm, 2024, doi:10.3390/jcm13216615_

Round 1

Reviewer 1 Report

Comments and Suggestions for Authors

Dear authors, 

thank you for the opportunity to read and review the manuscript. It is interesting and well written.

General comments

This retrospective analysis on a large asthmatic population is interesting, it highlights how personalized medicine could be the way to proper treat the disease and prevent its worsening.

Asthma disease has a burden on people health: early management and proper treatment could have an important role on quality of life and health status of these patients.

Specific comments

Introduction section is clear. The authors reported the pathophysiological features of the disease and the related phenotypes (also in the discussion section) with a reference to the different inflammatory pathways, I think that a brief description of the main pathways should be added, thus reflecting also the impact on different clusters that are reported in the results: Th2 high Th2 mediated with release and activation of (mostly) IL-3, IL-4, IL-13; Th2 low Th17 mediated, release and activation of IL-17, IL-17F, IL-21, IL 22 [D. Robinson,  M. Humbert,  R. Buhl,  A. C. Cruz,  H. Inoue,  S. Korom,  N. A. Hanania and  P. Nair, Clinical & Experimental Allergy,  2017 ( 47)  161175.)].

Table 1 is interesting, 

ACO is reported in the table (as well as in the discussion section). I agree that COPD and Asthma could coexist in the same patients, but this coexistence is no more defined as ACO syndrome as they are different disease (GOLD 2024).

Material and Method 

Study population: there were exclusion criteria? Pediatric patients were included or excluded? Figure 1, a) the age starts from 20 years old. Is it correct?  (line 499 please specify “the atopic allergy cluster refers mainly to the pediatric population, which is absent in our case due to the specific data collection”)

Results section is interesting and well written; temporal patterns, demographic factors, triggers and comorbidities were analyzed in this large populations distinguishing between J45 (asthma) and J46 (status asthmaticus).

Table 2, data regarding statistical description are reported, please add the number of patients in each group; as reported in discussion section the large difference in size should be reported also in the result section.

Discussion, line 270-276 even if I agree with the authors that patients with status asthmaticus require mechanical ventilation more often if compared to patients with asthma, the statistical significance of the test, considering the wide difference in number of patients in the two groups, should be taken cautiously.

The clusters founded by the authors are very interesting, furthers study are needed to validate these results and to personalize the treatment of these patients.

Author Response

Dear authors, 

thank you for the opportunity to read and review the manuscript. It is interesting and well written.

General comments

This retrospective analysis on a large asthmatic population is interesting, it highlights how personalized medicine could be the way to proper treat the disease and prevent its worsening.

Asthma disease has a burden on people health: early management and proper treatment could have an important role on quality of life and health status of these patients.

Specific comments

Introduction section is clear. The authors reported the pathophysiological features of the disease and the related phenotypes (also in the discussion section) with a reference to the different inflammatory pathways, I think that a brief description of the main pathways should be added, thus reflecting also the impact on different clusters that are reported in the results: Th2 highà Th2 mediated with release and activation of (mostly) IL-3, IL-4, IL-13; Th2 lowà Th17 mediated, release and activation of IL-17, IL-17F, IL-21, IL 22 [D. Robinson,  M. Humbert,  R. Buhl,  A. C. Cruz,  H. Inoue,  S. Korom,  N. A. Hanania and  P. Nair, Clinical & Experimental Allergy,  2017 ( 47)  161–175.)].

Table 1 is interesting, 

ACO is reported in the table (as well as in the discussion section). I agree that COPD and Asthma could coexist in the same patients, but this coexistence is no more defined as ACO syndrome as they are different disease (GOLD 2024).

Material and Method 

Study population: there were exclusion criteria? Pediatric patients were included or excluded? Figure 1, a) the age starts from 20 years old. Is it correct?  (line 499 please specify “the atopic allergy cluster refers mainly to the pediatric population, which is absent in our case due to the specific data collection”)

Results section is interesting and well written; temporal patterns, demographic factors, triggers and comorbidities were analyzed in this large populations distinguishing between J45 (asthma) and J46 (status asthmaticus).

Table 2, data regarding statistical description are reported, please add the number of patients in each group; as reported in discussion section the large difference in size should be reported also in the result section.

Discussion, line 270-276 even if I agree with the authors that patients with status asthmaticus require mechanical ventilation more often if compared to patients with asthma, the statistical significance of the test, considering the wide difference in number of patients in the two groups, should be taken cautiously.

The clusters founded by the authors are very interesting, furthers study are needed to validate these results and to personalize the treatment of these patients.

Reply to reviewer 1:

Thank you for your thoughtful and constructive review of our manuscript. We greatly appreciate the time and effort you have invested in providing detailed feedback, which will undoubtedly help improve the quality and clarity of our work.

Introduction:

Main pathways impact on different clusters – we added that discussion and highlighted the clinical significance.

ACO: thank you so much for bringing that to our attention. We added a discussion on the "Dutch hypothesis" and changed the first two cells in that row to reflect that.

Material and Method:

Study population: Added exclusion criteria

Results:

Table 2: Added the number of patients

Discussion:

Lines 270-276 changed them.

Thank you again for your constructive feedback. We believe these changes will improve our manuscript's quality and readability.

Reviewer 2 Report

Comments and Suggestions for Authors

• The chosen aim of the study is important. Probably, the study is done well. However, the information in the manuscript is very difficult to read. It needs a lot of corrections and shortened.

• The information provided in the Introduction in Lines 56-65 and Table 1 is redundant because does not meet the aim of the study.

• In lines 88-89, the authors stated, "All participants had previously provided written informed consent for their data to be 88 used for research purposes". How can that be in retrospect? Do all patients who enter the hospital sign a consent form in advance for all possible future research? I think that such a requirement is essentially redundant. For retrospective studies, patients' consent is usually not sought for publication when depersonalized data are used.

• The information presented in 3.3 and Figure 4 is not relevant for the international reader. This information does not add value.

• It is not entirely clear how the authors classified patients into phenotypes. In clinical practice, prospectively assigning a patient to a specific phenotype is quite problematic. In my opinion, it is completely impossible to do this in retrospect, especially over such a long period of 10 years. Especially when retrospectively classifying patients into Th2 and non-Th2 phenotypes, based only on medical records, which are usually not very comprehensive in a scientific sense.

• It is not very clearly written how patients were assigned to clusters and what the clinical meaning of this is.

• Part of the discussion is rather difficult to read, to follow the thought. This part should be rewritten, emphasizing not the statistical methods and relationships, but the clinical significance of the results.

• The conclusion of the study should be more specific, based on the results data.

• It is not clear whether the information presented in Lines 512-544 is the authors' reflections (i.e., a continuation of the Discussion but without references), a repetition of the results, or a conclusion.

Author Response

Comments and Suggestions for Authors

  • The chosen aim of the study is important. Probably, the study is done well. However, the information in the manuscript is very difficult to read. It needs a lot of corrections and shortened.
  • The information provided in the Introduction in Lines 56-65 and Table 1 is redundant because does not meet the aim of the study.
  • In lines 88-89, the authors stated, "All participants had previously provided written informed consent for their data to be 88 used for research purposes". How can that be in retrospect? Do all patients who enter the hospital sign a consent form in advance for all possible future research? I think that such a requirement is essentially redundant. For retrospective studies, patients' consent is usually not sought for publication when depersonalized data are used.
  • The information presented in 3.3 and Figure 4 is not relevant for the international reader. This information does not add value.
  • It is not entirely clear how the authors classified patients into phenotypes. In clinical practice, prospectively assigning a patient to a specific phenotype is quite problematic. In my opinion, it is completely impossible to do this in retrospect, especially over such a long period of 10 years. Especially when retrospectively classifying patients into Th2 and non-Th2 phenotypes, based only on medical records, which are usually not very comprehensive in a scientific sense.
  • It is not very clearly written how patients were assigned to clusters and what the clinical meaning of this is.
  • Part of the discussion is rather difficult to read, to follow the thought. This part should be rewritten, emphasizing not the statistical methods and relationships, but the clinical significance of the results.
  • The conclusion of the study should be more specific, based on the results data.
  • It is not clear whether the information presented in Lines 512-544 is the authors' reflections (i.e., a continuation of the Discussion but without references), a repetition of the results, or a conclusion.

Reply to Reviewer 2:

Thank you for your thorough review and valuable feedback on our manuscript. We appreciate your insights and will address each point to improve the clarity and relevance of our study. Here's our response to your comments:

Information is unclear: we added a paragraph in the introduction that describes how the paper is organized and how to read it if you’re more interested in the evidence-gathering process or the clinical applications.

Table 1 and its information: one of the aims of the study is to investigate the interactions between status astmaticus and comorbidities and how they reflect existing literature phenotypes, as stated in the introduction in question 4. Therefore, presenting the existing asthma phenotypes is part of the study.

Lines 88-89: yes, all patients sign an informed consent form upon admission since this is a University hospital and the data is used for teaching and research. We used the addresses of the selected lot, so the Ethics Committee considered it important for us to mention this. For clarification, we modified the sentence to add this aspect.

3.3. and Figure 4: The 2024 Global Initiative for Asthma (GINA) report emphasizes the importance of regional personalization in asthma management, recognizing that asthma care must be tailored to meet the unique needs of different populations and healthcare contexts. GINA acknowledges that asthma management strategies should be adapted to reflect local healthcare systems, resources, and the specific challenges faced by different regions. This includes considering socioeconomic factors, access to medications, and prevalent local environmental triggers for asthma. Furthermore, GINA provides a framework for developing national or regional guidelines that incorporate local evidence and practices. This customization is crucial for improving the effectiveness of asthma management globally. Therefore, we do not consider that the local informations are irrelevant to the international reader; on the contrary, they present a replicable methodology used to investigate the environmental triggers. We added a special clinical section describing these results in the discussion section to make them more actionable.

Unclear patient to phenotypes: the entire 3.4 section describes how we created the comorbidities network. To resume, we did not classify a specific patient into a category or the other but created a network that presents the interactions of all the comorbidities for all the patients, which is then mathematically split based on specific algorithms into clusters. Those clusters represent disease codes that are more connected to each other than to the other clusters. From a clinical standpoint, this means that they tend to be present together more frequently. Interpreting them, we discovered that some actually are very close to the literature phenotypes and realized that they are our local phenotypes.

Discussion unclear: we revised the Discussion section to emphasize the clinical implications of our results. We split it into numerical data interpretations and added a clinical meaning section to each result. We felt that by removing the statistical and numerical methods, we are removing essential components of evidence-based medicine, similar to both GINA and GOLD evidence-based reports, but we corralled those into specific stand-alone sections to be looked upon as needed.

Conclusions: we modified the paragraph so that it directly answers the original research questions and does not seem like a continuation of the previous section.

Round 2

Reviewer 1 Report

Comments and Suggestions for Authors

Dear authors,

thank you for the opportunity to read and review the revised version of the manuscript. Its quality has improved. It is interesting and well written.

Author Response

Dear Reviewer,

We sincerely appreciate your thorough and insightful review of our manuscript.

Your comments have been invaluable in enhancing the quality and clarity of our work.

Reviewer 2 Report

Comments and Suggestions for Authors

In my opinion, the manuscript is not well prepared for the publication – there is still a lot of redundant, unnecessary information.

My earlier concerns and recommendations remain.

Author Response

The authors thank the reviewer for their continued feedback and dedication to improving the quality of the work. We acknowledge the reviewer's concerns and appreciate their careful review of the manuscript. Therefore, the manuscript was thoroughly reviewed to identify and remove any redundant information. Each section was critically evaluated to ensure its relevance to the research questions and its contribution to the field. The presentation of data was streamlined to highlight the most impactful findings and their clinical significance, while keeping the research parts, relevant to the other reviewer.